# The Equal Neutralizing Effectiveness of BNT162b2, ChAdOx1 nCoV-19, and Sputnik V Vaccines in the Palestinian Population

**DOI:** 10.3390/vaccines12050493

**Published:** 2024-05-03

**Authors:** Alexia Damour, Muriel Faure, Nicolas Landrein, Jessica Ragues, Narda Ardah, Haneen Dhaidel, Marie-Edith Lafon, Harald Wodrich, Walid Basha

**Affiliations:** 1CNRS UMR 5234, Fundamental Microbiology and Pathogenicity, University Bordeaux, 33076 Bordeaux, France; alexia.damour@u-bordeaux.fr (A.D.); muriel.faure@u-bordeaux.fr (M.F.); nicolas.landrein@u-bordeaux.fr (N.L.); jessica.ragues@u-bordeaux.fr (J.R.); marie-edith.lafon@u-bordeaux.fr (M.-E.L.); 2IBGC, UMR 5095, CNRS UMR 5095, Institute of Cellular Biochemistry and Genetics, Université of Bordeaux, 33077 Bordeaux, France; narda.ardda@ibgc.cnrs.fr; 3Department of Biomedical Sciences, Faculty of Medicine and Health Sciences, An-Najah National University, Nablus P400, Palestine; 4Department of Applied and Allied Medical Sciences, Faculty of Medicine and Health Sciences, An-Najah National University, Nablus P400, Palestine; h.dhaidel@najah.edu; 5Virology Laboratory, Pellegrin Hospital, Bordeaux University Hospitals, 33076 Bordeaux, France

**Keywords:** SARS-CoV-2, vaccination, Palestine, neutralizing antibodies, mRNA vaccine, adenovirus vaccine

## Abstract

Since the beginning of the COVID-19 pandemic, different viral vector-based and mRNA vaccines directed against the SARS-CoV-2 “S” spike glycoprotein have been developed and have shown a good profile in terms of safety and efficacy. Nevertheless, an unbiased comparison of vaccination efficiency, including post-vaccination neutralizing activity, between the different vaccines remains largely unavailable. This study aimed to compare the efficacy of one mRNA (BNT162b2) and two non-replicating adenoviral vector vaccines (ChAdOx1 nCoV-19 and Sputnik V) in a cohort of 1120 vaccinated Palestinian individuals who received vaccines on an availability basis and which displayed a unique diversity of genetic characteristics. We assessed the level of anti-S antibodies and further determined the antibody neutralizing activity in 261 of those individuals vaccinated with BNT162b2a (121), ChAdOx1 (72) or Sputnik V (68). Our results showed no significant difference in the distribution of serum-neutralizing activity or S-antibody serum levels for the three groups of vaccines, proving equivalence in efficacy for the three vaccines under real-life conditions. In addition, none of the eight demographic parameters tested had an influence on vaccination efficacy. Regardless of the vaccine type, the vaccination campaign ultimately played a pivotal role in significantly reducing the morbidity and mortality associated with COVID-19 in Palestine.

## 1. Introduction

Since the initial 2019 outbreak of the novel beta coronavirus SARS-CoV-2, the global response has been characterized by the rapid development and deployment of various vaccines to control viral spread and protect individuals from the severe consequences of COVID-19. Alternative vaccine strategies have been developed to provide a safe and effective immune response to protect from severe disease [1]. These strategies include modified mRNA-based vaccines such as BNT162b2 (Pfizer-Biontech, New York, NY, USA/Mainz, Germany) and mRNA-1273 (Moderna, Cambridge, MA, USA) vaccines, non-replicating adenoviral vectors such as the ChAdOx1 nCoV-19 (AstraZenca/University of Oxford, Cambridge, UK/ Oxford, UK), Ad26.Cov2.S (Johnson & Johnson, New Brunswick, NJ, USA), Sputnik V and Sputnik light vaccines (Gamaleya Institute, Moscow, Russia) and inactivated viral vaccines such as BBIBP-CorV (Sinopharm, Shanghai, China) and CoronaVac (Sinovac, Beijing, China) [2,3,4]. Among these strategies, viral vector-based and modified mRNA vaccines have shown good profiles in terms of safety and efficacy [5].

The efficacy of these vaccines varied, with reported rates of protection between 70% and 95% in the respective phase III clinical trials [6,7,8,9,10,11,12]. The initial efficacy trials did not involve direct comparisons between different vaccines, and the study design and country-specific biased vaccination strategies have made an impartial comparison of vaccination efficiency between the different available vaccines problematic. An important indicator of the development of an immune response in vaccinated individuals is the level of neutralizing antibodies [13]. Such antibodies have been detected in varying levels in individuals who were vaccinated with different vaccines and were shown to be correlated with protection against symptomatic infection [13]. The complex interactions between vaccine types, host genetics and environmental factors may influence the outcomes of vaccination in ways that are not yet fully understood [14].

The vaccination campaign in the Palestinian Occupied Territory (POT) was ambitious, with more than 3.7 million vaccines administered as of 17 October 2023 [15]. These vaccines included, for the most part, BNT162b2, ChAdOx1 Sputnik V and, to a lesser extent, mRNA-1273, BBIBP-CorV and CoronaVac [16]. The variety in vaccine types, each proposed according to availability during the vaccination campaign, as well as the number of vaccinated individuals and the genetic diversity characteristic for the Palestinian population create a fertile ground for investigating the development of neutralizing antibodies by different vaccines and the effect they had on the progression of the disease.

The present study had two main objectives: (1) to assess the antibody response provoked by the main three vaccine types administered to the Palestinian population (BNT162b2, ChAdOx1, Sputnik V) and (2) to compare retrospectively the serum-neutralizing activity produced by each of these three vaccines. Through these objectives, this research aimed to provide comprehensive insights into the effectiveness of the vaccination campaign within the Palestinian population, taking into account several demographic parameters, the use of different vaccine types and the associated neutralizing activity of the elicited antibody response.

## 2. Materials and Methods

### 2.1. Study Design and Settings

A cross-sectional study was conducted among Palestinians from the West Bank who received the COVID-19 vaccine prior to 15 May 2022. The inclusion criteria were people aged between 18 and 80 years old who received two doses of the BNT162b2 vaccine or completed the two-shot regimen of the Sputnik V or ChAdOx1 vaccines. Participants over 80 or under 18 years old with autoimmune diseases or immunocompromised conditions who received the Sputnik Light vaccine or received only one shot of the Sputnik V or ChAdOx1 vaccine were excluded from the study.

### 2.2. Blood Sampling and Data Collection

Approximately 5 mL of venous blood was collected from each participant in a two-month time window between 6 and 8 months following the last vaccine dose, with subsequent separation of serum from the blood followed by storage at −80 °C until required. In conjunction with the blood samples, the participants were required to fill out a questionnaire encompassing diverse demographic and clinical factors.

### 2.3. Cell Lines

The cell-based neutralization assay was described previously [17]. Briefly, the assay used syncytia formation between two modified human bone osteosarcoma epithelial cell lines (U2OS, ATCC HTB-96, generously provided by M. Piechaczyk, IGMM, Montpellier, France). The cells were genetically labeled with either GFP or mCherry, with GFP expressing U2OS cells expressed the SARS-CoV-2 receptor Angiotensin Converting Enzyme-2 following lentiviral transduction (ACE2, addgene #145839) and mCherry expressing U2OS cells which expressed all four codon-optimized SARS-CoV-2 structural proteins from the Wuhan strain: nucleoprotein (N, addgene #141391), membrane protein (M, based on addgene #141274), envelope protein (E, based on addgene #141273), and spike surface protein (S, based on addgene #149329). The resulting cell lines U2OS-GFP-ACE2 and U2OS-mCherry-NEMS were cultured in Dulbecco’s Eagle Medium (DMEM, Gibco, Gaithersburg, MD, USA) supplemented with 10% fetal bovine serum (FBS, Gibco) and 1% penicillin-streptomycin (Gibco), and maintained at 37 °C in a 5% CO_2_ humidified atmosphere. Upon reaching confluence, the cells were washed with phosphate-buffered saline (PBS, Gibco) and detached using 0.05% trypsin-EDTA (Gibco).

### 2.4. Neutralization Assay

U2OS-GFP-ACE2 and U2OS-mCherry-NEMS were combined at a 1:1 (*v:v*) ratio and subsequently seeded at a density of 5 × 10^4^ cells in 150 µL DMEM supplemented with 10% FBS and 1% penicillin-streptomycin per well in 96 well black wall plates (Ibidi, Gräfelfing, Germany). Fifty µL of serum from vaccinated individuals were added to each first well of a row and sequentially diluted in 1:4 steps. The cells were then cultured for 24 h to facilitate the formation of syncytia between the two cell lines in the presence of the serum dilution. The samples from individuals were numbered and blinded prior to the assay. In each assay, non-syncytia-forming controls (U2OS-GFP) were included and treated similarly. After 24 h, each well was captured at a 2.5× magnification using a fully automated CellDiscoverer-7 microscope system (Zeiss, Jena, Germany). Neutralization was assessed by calculating the mean GFP cell surface area using CellprofilerTM software (2.2.1), representing the formed syncytia surface. Ultimately, the serum neutralization titer was determined to be the IC50 representing half-syncytia inhibition based on values obtained from the serial dilution using Prism7.

### 2.5. SARS-CoV-2 CLIA Assay

All serum samples were anonymized and tested for the total anti-S antibodies using CLIA (SNIBE, Maglumi SARS-CoV-2 antigen). According to the manufacturer’s recommendations, samples were considered positive above a cutoff index of 1 AU/mL. All samples with values over 100 AU/mL were diluted and measured as 1/10 or 1/20, allowing extension of the dynamic range of analysis to 2000 AU/mL. For representation, all AUs were converted into the WHO standard *binding antibody units* (BAUs) using the recommended multiplication factor of 4.33 [18].

### 2.6. Statistical Analysis

The calculation of the required sample size to meet the objectives of the research and ensure sufficient statistical power was based on the equation *n* = [DEFF × Np(1 − p)]/[(d^2^/Z^2^_1−α/2_ × (N − 1) + p × (1 − p)]. To cover different geographical areas in Palestine, the West Bank was divided into three regions: north, middle, and south. The sample size was calculated for each region. The calculated sample size for each of the West Bank regions was 370. Therefore, the minimum acceptable total sample size was 1110. Multiple data comparison was performed using ordinary one-way ANOVA with a Tukey multiple comparison test. A comparison of two parameters was performed using an unpaired parametric two-tailed *t*-test. To assess the significance of the correlation between two measured variables (e.g., age-IC50), the Pearson correlation coefficient was calculated. Significance levels were denoted as follows: * = *p* < 0.05, ** = *p* < 0.01, *** = *p* < 0.001, and **** = *p* < 0.0001, while “ns” indicated non-significance. All raw data points are graphically presented, and statistical analyses were computed using PRISM 7.

### 2.7. Ethical Approval, Registration and Patient Consent Procedures

All procedures conducted in this study adhered to both federal and institutional ethical guidelines, the 1964 Helsinki Declaration and subsequent amendments or equivalent ethical standards. Approval for the study was granted by the Institutional Review Board (IRB) committee of An-Najah National University (Reference No. Med Nov.2021/32) and the Palestinian Ministry of Health. Participants were invited to voluntarily take part in the study, and those who chose to participate provided informed consent by signing a consent form. The study’s background and objectives were thoroughly explained to the participants to ensure their understanding.

### 2.8. Data Availability

The authors confirm that the data used for the findings in this study will be made available through the corresponding authors to qualified and interested investigators upon reasonable request.

## 3. Results

### 3.1. Demographic Characteristics of the Participants

In this study, the data were collected from 1120 participants. The mean age of the participants was 32.0 ± 14.8 years, and the mean body mass index (BMI) was 24.9 ± 4.6 kg/m^2^. Among the participants, 769 individuals (68.7%) were below the age of 40, 540 participants (48.2%) were female, 53 and 537 individuals (4.7% and 47.9%) fell into the underweight or normal weight category, respectively, and 788 participants (70.4%) identified as non-smokers. In terms of vaccine distribution, 727 participants (64.9%) received the Pfizer-BioNTech vaccine, 185 participants (16.5%) were administered the AstraZeneca vaccine, and 208 participants (18.6%) received the Sputnik V vaccine. Additionally, individuals with blood types A and O constituted 75.2% of the participants, with 36.0% and 39.2% representing each blood type, respectively (Table 1).

### 3.2. Post-Vaccine SARS-CoV-2 Total Antibodies

Out of 1120 vaccinated participants, only 11 tested negative for SARS-CoV-2 specific S antibodies. The study found no significant differences in the serum levels of specific S antibodies according to vaccine type or other demographic factors, indicating a consistent trend across the participant cohort (*p* > 0.05) (Table 1).

### 3.3. Serum Neutralization among Individuals Vaccinated with BNT162b2, ChAdOx1 or Sputnik V

The vaccination campaign for the Palestinian population distributed vaccine doses on an availability basis. The mRNA-based vaccine BNT162b2 and the two adenovirus vector-based vaccines ChAdOx1 and Sputnik V were the most applied vaccines. Amongst the cohort of 1120 individuals, we thus randomly selected 261 sera from people who had received at least two doses of a single type of vaccine. Of those, 121 individuals had been vaccinated with BNT162b2a, 72 with ChAdOx1 and 68 with Sputnik V. We then assessed the serum neutralization capacity using our in-house syncytia fusion assay. We determined the syncytia inhibition titer (IC50) for all individuals as a measure of serum neutralization. For the assay, the samples were numbered and analyzed in a double-blind protocol before the sample identity was revealed for further analysis. We first analyzed the distribution of the calculated IC50 values for each vaccine type (Figure 1A). We observed no significant difference in the distribution of IC50 values between the three different vaccine types. Within each vaccinated group, we found individuals with high neutralization and low neutralization activity (Figure 1B). For ethical reasons, no non-vaccinated control group was included in our assay, but we considered IC50 neutralizations values below 25 to be low responders based on our previous application of the assay [17]. Using thresholds, we also did not observe any significant difference in the neutralization efficiency between the three vaccine types, as our analysis was restricted to low responders (IC50 < 25) or high responders (IC50 > 250). The correlation between the level of S-antigen-specific antibodies in the serum and the serum neutralization capacity were analyzed next. Consistent with previous studies showing a decline of such a correlation over time [17,19], we observed a poor overall correlation between the calculated IC50 values and the anti-S antibody levels (Figure 1C). This was even clearer with a pairwise comparison of the S-antibody level (in BAUs) and IC50 neutralization titer per individual (Figure 1D). Interestingly, sera from three vaccinated individuals without detectable S-antibodies (marked with * in Figure 1D) were included in the analysis, and two of them had low but detectable neutralization activity.

### 3.4. Correlation between Serum Neutralization Efficacy and Demographic Parameters across BNT162b2, ChAdOx1 or Sputnik V Vaccinated Individuals

In each vaccinated group, we observed that part of the vaccinated individuals were low responders concerning the neutralization efficacy. In general, the individual response to vaccination is influenced by many parameters, of which the vaccine type is just one [14]. Other parameters were therefore investigated. For the analysis, the data for all three vaccines were pooled (Figure 2). First, the impact of intrinsic host factors on neutralization response was assessed, especially since several reports indicated a reduced immune response in the elderly [20] as well as a gender-specific better antibody response for several vaccines (summarized in [14]). Our analysis showed neither correlation between neutralization efficacy and age (Figure 2A) or height (Figure 2B) nor differences when comparing the two genders (Figure 2C). There also was no correlation between IC50 and weight (Figure 2D) or BMI (Figure 2E), although a negative correlation between BMI and antibody response was reported for certain vaccine types [21], including for SARS-CoV-2 vaccines [22]. Blood groups, which have been reported to affect the humoral and cellular response following the application of an oral cholera vaccine [23,24], were another intrinsic host factor without influence on the IC50 in our analysis (Figure 2F). We next assessed the environmental and behavioral factors that were collected within the cohort and which may affect vaccine efficacy outcomes due to increased potential exposure to other pathogens [25]. Comparing the individuals living in urban, rural or camp settings did not explain the differences in the IC50 responses, although the latter sample size was quite small (Figure 2G). Likewise, we did not identify a relation between the smoking status of individuals and their IC50 responses (Figure 2H). Each part of the analysis was also performed individually for each vaccine type and showed no significant differences [26], thus further confirming the observed equivalence of the three vaccines. Taken together, none of the parameters we analyzed were sufficient to explain the difference in the neutralization response within the vaccinated population.

## 4. Discussion

Following the outbreak of the COVID-19 health crisis, the Palestinian Ministry of Health and the WHO started a vaccination campaign throughout the POT in the late summer of 2021, with a peak in activity into early 2022. The campaign included multiple vaccines on an availability basis. Due to early availability, the Sputnik V and Sinopharm vaccines were initially administered, while Pfizer-BioNTech and Astra-Zeneca became available later through the COVAX fund, with the final shipment in December 2021 also including Moderna [16]. Here, we analyzed the effectiveness of the vaccination campaign by determining the elicited anti-S antibody levels in a random cohort of 1120 individuals who received either the Pfizer-BioNTech, AstraZeneca or the Sputnik V vaccine and for whom serum was collected 6–8 months post vaccination (see Table 1). We further determined the IC50 value for the serum neutralization activity in a part of the cohort. Importantly, the use of a diverse range of vaccine types based on availability, a peak period of vaccine application in late 2021 and the collection of serum within 6–8 months following application of the last vaccine dose retroactively created a unique situation to fairly assess the effectiveness of individual vaccines in this single cohort.

Above all, our results demonstrate the success of the Palestinian vaccination campaign, with almost every tested individual showing significant levels of S-antibodies and active serum neutralization activity. However, the cohort only included fully vaccinated individuals and blended out the supply dependence on COVAX and other countries for vaccine donations, which resulted in a delay for the vaccination campaign and a vaccine uptake disparity [16].

Our analysis also demonstrated that neither the level of elicited anti-S antibodies (in BAUs, Figure 1) nor the neutralization capacity of the serum (IC50, Figure 1) was linked to the type of vaccine included in this study. Both the S-antibody levels and serum neutralization activity were distributed over a large range and poorly correlated. Only 11 individuals tested negative for S-antibodies, of which three were tested further for neutralization, with two showing low but measurable serum neutralization activity. This outcome of vaccine equivalence is also supported through meta-data analysis showing that vaccination with several vaccines, including all three vaccines from this study, results in protection from severe COVID-19 disease [27]. There is some inconsistency, with available studies directly comparing vaccine effectiveness between mRNA- and vector-based vaccines revealing somewhat reduced serum neutralization activity in ChAdOx1-vaccinated individuals compared with BNT162b2, especially with variants [28,29]. The reason for this difference could be due to the non-standardized and late collection time of serum post vaccination in our study. Most comparative studies measured the serum effects within days or weeks after vaccination completion [29,30,31]. This may be an important factor, as when applying the same assay, we recently followed an unrelated cohort receiving mRNA-based vaccination over time, showing that the antibody and neutralization levels declined over time, as well as the number of applied vaccine doses [17]. Indeed, a recent study comparing mRNA- and Ad vector-based vaccines over time showed that mRNA- but not Ad-based vaccines induce a rapid but short-lived peak in S-specific antibodies in the serum which levels out over time [32], potentially dropping to levels with non-significant differences between vaccine groups [32,33]. As a consequence, initial differences in the vaccine response may be less important for the long-term establishment of protective immunity, as shown in our study. We also tested several demographic parameters for their influence on the vaccination response but were unable to find any correlation with the vaccine effectiveness (Figure 2). This result further indicates that individual vaccine responses are multifactorial [14].

One shortcoming of our study is that we did not account for circulating SARS-CoV-2 variants, nor did we measure cellular immunity or attempt MHC characterization in the participants due to the available local setting. Nevertheless, our study is meaningful because it was performed on a large cohort of vaccinated individuals without a biased strategy and shows the success of the vaccination campaign. Notably, it retroactively qualifies the use of more cost-effective vaccines and suggests equivalence between mRNA- and adenovirus vector-based COVID-19 vaccines when applied consistently under field conditions. By highlighting the efficacy of the vaccination program in the POT, this study emphasizes the importance of providing vaccines for all communities, including Palestinians. Moving forward, continued research efforts and collaboration will be essential in ensuring and evaluating the ongoing success of vaccination initiatives and in safeguarding the health and well-being of all communities.

## Figures and Tables

**Figure 1 vaccines-12-00493-f001:**
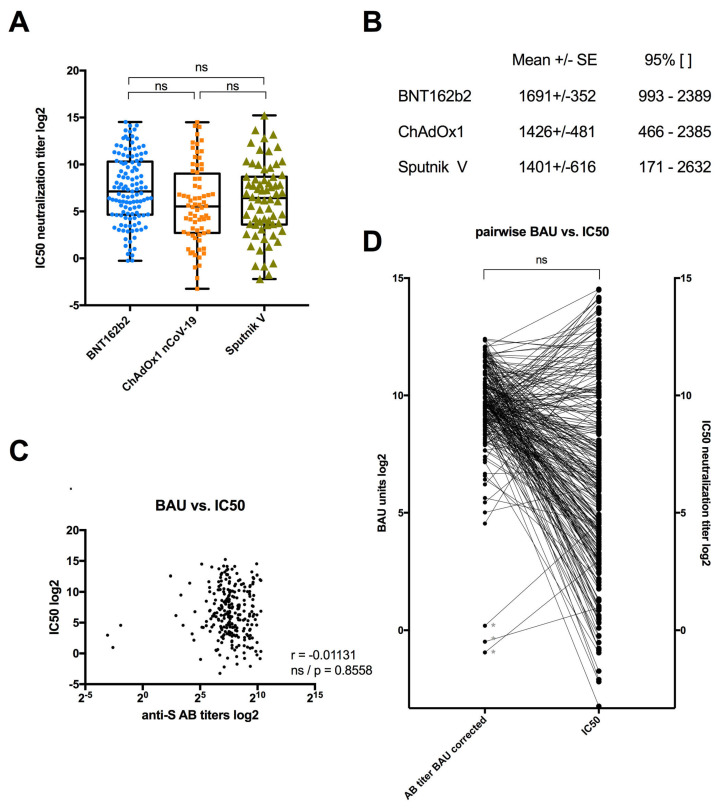
Serum neutralization following vaccination with BNT162b2, ChAdOx1 nCoV-19 or Sputnik V. (**A**) Serum neutralization was determined based on syncytia inhibition (IC50) for vaccinated patients with BNT162b2 (blue, *n* = 121), with ChadOx1 nCoV-19 (orange, *n* = 72) or with Sputnik V (green, *n* = 68). One-way ANOVA with a Tukey multiple comparison test (ns = not significant). (**B**) Mean of IC50 +/− standard error and 95% confidence interval for each type of vaccine tested. (**C**) BAU levels (anti-S Ab titers) and IC50 for each vaccinated patient were normalized and plotted. Pearson correlation coefficient (r) as indicated (ns = not significant). (**D**) Pairwise comparison of BAU levels (anti-S Ab titers) and IC50 for each patient and paired two-tailed *t*-test (ns = non-significant). Gray * indicates an individual who tested negative for S-antigen.

**Figure 2 vaccines-12-00493-f002:**
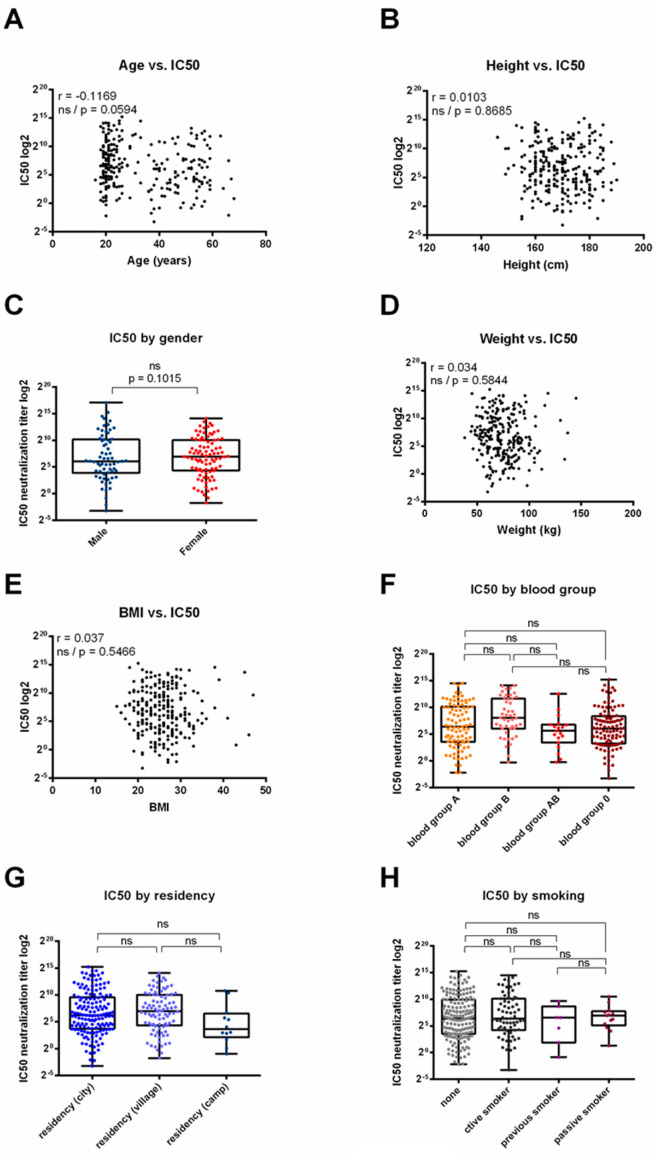
Correlation between serum neutralization and demographic parameters in patients vaccinated with BNT162b2, ChAdOx1 nCoV-19 or Sputnik. (**A**,**B**,**D**,**E**) IC50 and age (**A**), height (**B**), weight (**D**) and BMI (**E**) for each vaccinated patient were normalized and plotted. Pearson correlation coefficient (r) as indicated (ns = not significant). (**C**,**F**,**G**,**H**) Serum neutralization was determined based on syncytia inhibition (IC50) for vaccinated patients according to their gender (**C**), blood group (**F**), residency (**G**) or smoking status (**H**). One-way ANOVA with a Tukey multiple comparison test (ns = not significant).

**Table 1 vaccines-12-00493-t001:** Demographic characteristics of the participants and their correlation with the total antibody titer. Distribution of demographic parameters and the respective levels of SARS-CoV-2 spike (S) specific antibody titer in the analyzed cohort (see text for details).

	Frequency	Percent	Mean of Antibody Titer	*p* Value
			(AU/mL)	
Age				
18–39	769	68.7	285.5	0.591
40–49	119	10.6	294.8	
50 and above	232	20.7	372.0	
Body Mass Index (BMI)				
Underweight	53	4.7	245.4	0.602
Normal	537	47.9	316.7	
Overweight	384	34.3	306.8	
Obese	146	13.0	276.5	
Gender				
Male	580	51.8	332.3	0.105
Female	540	48.2	274.9	
Smoking				
Non-Smoker	788	70.4	313.7	0.087
Current Smoker	332	29.6	283.0	
Blood Group				
A	403	36.0	321.2	0.072
B	190	17.0	322.7	
AB	88	7.9	312.1	
O	439	39.2	280.1	
Type of Vaccine				
Pfizer	727	64.9	322.1	0.126
AstraZeneca	185	16.5	227.1	
Sputnik V	208	18.6	312.8	

## Data Availability

The authors confirm that the data used for the findings in this study will be made available through the corresponding authors to qualified and interested investigators upon reasonable request.

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
