# Peer review of "The Equal Neutralizing Effectiveness of BNT162b2, ChAdOx1 nCoV-19, and Sputnik V Vaccines in the Palestinian Population"

_vaccines, 2024, doi:10.3390/vaccines12050493_

Round 1
Reviewer 1 Report (Previous Reviewer 1)
Comments and Suggestions for Authors
All requested changes have been made and issues answered correctly
Reviewer 2 Report (Previous Reviewer 2)
Comments and Suggestions for Authors
The authors have improved the manuscript by suitably responding to all my critiques.
Reviewer 3 Report (Previous Reviewer 3)
Comments and Suggestions for Authors
The authors made changes to the manuscript according to the recommendations of the reviewers.
This manuscript is a resubmission of an earlier submission. The following is a list of the peer review reports and author responses from that submission.
Round 1
Reviewer 1 Report
Comments and Suggestions for Authors
Only minor editing of English language are required
Author Response
We would like to thank the editor and the reviewers for their positive reception and the time spend on our manuscript. We also appreciate the helpful comments and suggestions to which we respond in a point-by-point reply below. In addition, we modified both figures and re-analysed certain data in line with the suggestions of the reviewers detailed below. We also gave the manuscript to a native English-speaking colleague for revision.
Reviewer 1
The article, written by Damour et al., aims to compare the immune response, particularly the neutralizing response against SARS-CoV-2 virus in subjects vaccinated with BMT162bs, ChAdOx1 and Sputnik V. The work is clear, and the results are consistent with the intended purpose. A strong point is certainly the large number of subjects analysed.
We thank the reviewer for this positive feedback.
However, I have a few comments and revisions to suggest:
- Major revisions
- Some references are to be implemented or added, e.g. lines 42-45, 50-54
We added more key references to the background section and the discussion.
- Paragraph 2.1: Is information available for each group on the average time (with SE) since vaccination? Although specified in line 251, adding this specific information would greatly support the results, as line 274-276 suggest that these criteria might not have been met.
For all study participants (including for all three vaccines) serum samples were collected in a 2 month time window, at least 6 months after concluding the vaccination regimen and not later than 8 months. We have clarified this information now in the text (line 88). Unfortunately, the exact dates for sample collection per each individual is not available preventing the calculation of average and SE but there was no bias in serum collection between the three vaccines.
- Performing neutralization assay against circulating variants of SARS-CoV-2 would add considerable value to the work, as the response against Wuhan strain of the three vaccines has already been extensively characterised in other works, both individually and in comparison.
We agree with the reviewer that it would be great to have this information in general but also as reassurance for the Palestinian population. There are two main reasons why we decided not to include this analysis. Firstly, we found it challenging to identify circulating variants in time because SARS-CoV-2 consistently evolves and without support of timely sequencing data (i.e. not available for the Palestinian population) it is difficult to decide which variant will become predominant in Palestine. Independent of this study, we updated our neutralization cell-cell fusion assay using the “S” protein from the omicron variant because omicron (and its variations) became the dominant global variant. However, one of the main differences between the original SARS-CoV-2 variants and the Omicron variant is that the latter enter cells by endocytosis and fuse from within the endosome presumably after Spike processing through lysosomal cathepsins (PMID: 35798890, PMID: 37243215) unlike the original strain which infects cells by fusion at the plasma membrane following processing of the spike protein through TMPRSS-2 protease (PMID: 34159616). As such our assay with the updated omicron spike did not produce sufficiently large syncytia to serve as reliable neutralization assay.
Still we think that our study will considerably contribute. We did find only a limited number of comparative studies with several vaccines especially rarely under real field conditions. Thus, the comparative nature of our study performed under real life conditions with a large number of individuals and circumstantially unbiased will provide additional and valuable data to the interested research community. In addition, it represents an important feedback for the Palestinian population.
- Minor revisions
- Line 44: Protection from severe disease
We changed the text accordingly
- Paragraph 2.1: Is information available for each group on the average time (with SE) since vaccination? Although specified in line 251, adding this specific information would greatly support the results, as line 274-276 suggest that these criteria might not have been met.
(please see comment above)
- Line 121: The anti-S antibody (Maglumi SARS-CoV-2 antigen) assay was performed by ELISA or CLIA? I can only find references as a CLIA method for this assay.
The reviewer is correct. The technology used is CLIA. We changed the text.
- Line 108: it would be better to change “50 µl” to “Fifty µl” as it is at the end of the sentence.
We changed the text accordingly
- Table 1: the age group in the centre (40-49) is very narrow compared to the others> Were there any specific criteria behind this selection?
We thank the reviewer for this observation. Our main purpose was to define the “old” age group as “over 50” because many studies refer to this age as cutting point to define age groups (e.g. PMID:35202601). In return, we defined “young” as “under 40” leaving a narrow in between age group as the reviewer pointed out. To reply to this remark, we reanalysed the data changing the age group but again did not observe any significant difference between alternative age intervals. The apparent increase in antibody levels for “old age” in table 1 is statistically not significant due to a high SD within the group (see also comment 3, reviewer 2). We thus left table 1 in its current form.
- Line 175: Please specify the number of vaccine doses administered (I guess two, but it is better to specify).
All study participant received at least two doses of their respective vaccine type. This was an inclusion criteria for this study. This is now stated specifically in line 81 (material and methods) and again in line 177 as pointed out by the reviewer.
Reviewer 2 Report
Comments and Suggestions for Authors
This study compared the efficacy of three popular Covid-19 vaccines in the Palestinian population by assessing the level of S antibody and antibody neutralizing activity. It is a straightforward study and the manuscript is clear. Although it is not a very complete investigation, the new data will provide some good supplemental information for relevant researchers. However, there are some concerns to be addressed.
1. Line 23, 181 and 187: please add “significant” before “difference”.
2. Line 77: To my knowledge, BNT162b2 is a two-shot vaccine, but in this statement, it looks only one-shot.
3. Table 1: mean of antibody titer of age 50 and above is substantially higher than other age groups, so if you compare this way, the difference may be significant. It is hard to believe that no any positive difference throughout this whole study, since many other similar studies found some significant difference.
4. I don’t understand why the authors randomly selected 261 sera to do neutralization test from the cohort of 1120 individuals. What was the selection standard? Why didn’t you do serum neutralization for the whole cohort? I believe that should be better.
5. Two sera had no detectable S-antibodies but had detectable neutralization activity, how do you explain this result? It sounds not reasonable.
6. As the authors discussed, non-standardized collection time of serum post vaccination could be a major concern in this study. 6-8 months mean a possible 2-month difference, which is a long time and could lead to some false results.
Author Response
Reviewer 2
Comments and Suggestions for Authors
This study compared the efficacy of three popular Covid-19 vaccines in the Palestinian population by assessing the level of S antibody and antibody neutralizing activity. It is a straightforward study and the manuscript is clear. Although it is not a very complete investigation, the new data will provide some good supplemental information for relevant researchers. However, there are some concerns to be addressed.
We thank the reviewer for his positive reception of our study.
- Line 23, 181 and 187: please add “significant” before “difference”.
We have incorporated significant into the text.
- Line 77: To my knowledge, BNT162b2 is a two-shot vaccine, but in this statement, it looks only one-shot.
All study participants have received at least two vaccine shots (including BNT162b2). We added this information to the text in line 81 and 177 (please see also comment 9, reviewer 1).
- Table 1: mean of antibody titer of age 50 and above is substantially higher than other age groups, so if you compare this way, the difference may be significant. It is hard to believe that no any positive difference throughout this whole study, since many other similar studies found some significant difference.
The reviewer observation is correct. However, the apparent increase in antibody levels for study participants above the age of 50 in table 1 is statistically not significant due to a high SD within the group (see also comment 3, reviewer 2). A similar observation was done in (PMID: 35396570) showing an increase in the SD of antibody serum levels in older individuals compared to younger without overall statistical significance. We thus left table 1 in its current form.
Concerning the second comment, we agree with the reviewer that most published comparative studies found slight differences between different vaccine types. We discussed this in our original manuscript (line 269-276). We collected samples between 6 and 8 month post final vaccination. One similarity of those studies finding differences (mostly in favour of mRNA) is that antibody responses and neutralization efficiency was measured shortly after the vaccination protocol was completed (e.g. 7 or 28 days in PMID: 34883053, 14 days in PMID: 37903679, or ~90 days in PMID: 35350781). Indeed, a prospective comparative nationwide study in the Netherlands population compared S-specific antibodies elicited by 4 different vaccines over time (mRNA = Comirnaty, Spikevax and Ad vector based = Vaxzeria, Janssen) showing that mRNA based vaccines elicit a more rapid and higher induction of antibody serum levels (compared to Ad vector based vaccines) followed by a rapid decay, which was not observed for the Ad based vaccines. The study further shows that with time progression (here up to two months) individual antibody levels become more variable (higher SD) and initial existing significant differences between vaccine types disappear (PMID: 35396570). As another example one study measured antibody levels and neutralization at much later times (e.g ~8 months post vaccination in PMID: 36044963) describing no significant difference between BMT162bs vs. ChAdOx1. Thus, our results with serum taken 6-8 month post vaccination are in agreement with the literature, confirming that differences between vaccines are observed during the initial response to the vaccine but are less important for the long term establishment of protective immunity. To clarify this point we modified and expanded our discussion (line 272-282)
- I don’t understand why the authors randomly selected 261 sera to do neutralization test from the cohort of 1120 individuals. What was the selection standard? Why didn’t you do serum neutralization for the whole cohort? I believe that should be better.
This decision was based on logistic and practical considerations. S-specific antibodies were measured in Palestine, the neutralization tests were done in Bordeaux. For this purpose serum samples had to be shipped from the POT to France, which poses administrative and logistical challenges. To minimize the risk of loss or damage (hold up at customs, border crossings etc..) of the samples we decided to only analyse a part of the sera for neutralization. Sera for the neutralization test were sampled at random from each of the three vaccine groups retaining the overall proportion of vaccine group distribution in the cohort.
- Two sera had no detectable S-antibodies but had detectable neutralization activity, how do you explain this result? It sounds not reasonable.
The reviewer makes an important point. We re-analyzed the three negative samples using a different test (Abbott) also testing for N to exclude that our initial test was not sensitive enough or that neutralization comes from an undetected infection status. However, all three samples remained S- (and N-) negative. We also re-analyzed the images for the neutralization test. While the lower of the two S-negative neutralizing samples is within the detection accuracy of our neutralization test, re-examining the second S-negative sample clearly identified week but measurable neutralization activity. At this point we can only speculate that the serum contains S-specific neutralizing antibodies that are not picked up by either of the two commercial tests we used or that the levels are below the detection level of the respective tests.
- As the authors discussed, non-standardized collection time of serum post vaccination could be a major concern in this study. 6-8 months mean a possible 2-month difference, which is a long time and could lead to some false results.
We understand that this reviewer is concerned with the absence of significant differences in our study due to the time difference between vaccination completion and serum collection. We however, disagree with the interpretation that these results are therefore “false”. As detailed in our response to comment three above we think that our data are in agreement with previous studies and show the expected result for the long-term vaccination outcome.
Reviewer 3 Report
Comments and Suggestions for Authors
The authors presented the results of a cross-sectional study assessing the immune status of Palestinians vaccinated with three different vaccines against COVID-19 and the level of virus-neutralizing antibodies induced by these vaccines. A large amount of research has been carried out. However, there are some recommendations to improve the presented research results.
To fully assess the effectiveness of vaccines, it is optimal to provide data on the incidence of study participants after vaccination.
The authors talk about assessing the effectiveness of the vaccination campaign. It is recommended to justify the number of individuals (1120 people). What criteria were used to calculate the number of individuals?
In addition, it is recommended to bring the symbols of divisions on the graph axes to a single form (Figure 1). When using log2 units, the values on the scale should not be 210 (for example), but "10".
Author Response
Comments and Suggestions for Authors
The authors presented the results of a cross-sectional study assessing the immune status of Palestinians vaccinated with three different vaccines against COVID-19 and the level of virus-neutralizing antibodies induced by these vaccines. A large amount of research has been carried out. However, there are some recommendations to improve the presented research results.
We thank the reviewer for appreciating the work we put into our study.
To fully assess the effectiveness of vaccines, it is optimal to provide data on the incidence of study participants after vaccination.
We agree with reviewer that these would be useful data to present. Unfortunately, we do not have reliable data because they were collected as part of the questionnaire at 6 month post vaccination jointly with the blood sample. Study participants unfortunately couldn't give exact answers of infection incidents (e.g. confusing flu and covid-19) and many individuals avoid being tested for personal reasons.
The authors talk about assessing the effectiveness of the vaccination campaign. It is recommended to justify the number of individuals (1120 people). What criteria were used to calculate the number of individuals?
The calculation of the required sample size to meet the objectives of the research and ensure sufficient statistical power was based on the equation: n = [DEFF*Np(1-p)]/ [(d2/Z21-α/2*(N-1)+p*(1-p)]. To cover different geographical areas in Palestine, the West Bank was divided into three regions: the north, middle, and south; the sample size was calculated for each region. The calculated sample size for each of the West Bank regions was 370. Therefore, the minimum acceptable total sample size was 1110. This information is now included in the material and method section in line (129-134)
In addition, it is recommended to bring the symbols of divisions on the graph axes to a single form (Figure 1). When using log2 units, the values on the scale should not be 210 (for example), but "10".